# Predictors of Overnight and Emergency Treatment among Homeless Adults

**DOI:** 10.3390/ijerph17124271

**Published:** 2020-06-15

**Authors:** Chisom N. Iwundu, Pooja Agrawal, Michael S. Businelle, Darla E. Kendzor, Lorraine R. Reitzel

**Affiliations:** 1Department of Rehabilitation and Health Services, College of Public and Health, University of North Texas Services, Denton, TX 76203, USA; 2Department of Psychological, Health, and Learning Sciences, College of Education, University of Houston, Houston, TX 77204, USA; poagrawa@utmb.edu (P.A.); lrreitze@central.uh.edu (L.R.R.); 3School of Medicine, University of Texas Medical Branch, Galveston, TX 78701, USA; 4Oklahoma Tobacco Research Center, The University of Oklahoma Health Sciences Center, 655 Research Parkway, Suite 400, Oklahoma City, OK 73104, USA; michael-businelle@ouhsc.edu (M.S.B.); darla-kendzor@ouhsc.edu (D.E.K.); 5HEALTH Research Institute, University of Houston, 4849 Calhoun Rd., Houston, TX 77204, USA

**Keywords:** homeless, emergency treatment, hospitalization, sociodemographic, sex

## Abstract

High usage of emergency treatment and hospitalization has been reported among homeless individuals. Hence, this study aimed to identify the sociodemographic predictors associated with overnight and emergency hospital treatment among a sample of homeless adults. Participants were recruited from a shelter in Dallas, Texas (N = 354; Mage = 43.7 ± 11.7) and were predominantly uninsured, low-income men from various racial groups. The outcome variables were: (a) stayed overnight for treatment in a hospital; and (b) treated in a hospital emergency room. In logistic regression models, sex emerged as the only predictor of overnight treatment in a hospital (OR = 2.68, 95% CI = 1.61–4.47), and treatment in an emergency room (OR = 2.21, 95% CI = 1.34–3.65), such that women were more likely than men to be treated overnight and use emergency care. Targeted interventions and policies are needed to address homeless women’s primary care needs and reduce costlier treatment.

## 1. Introduction

On any given night, over half of a million individuals experience homelessness in the United States [1]. Homeless individuals experience high rates of disease burden in addition to dealing with various competing problems on a daily basis [2,3,4]. For example, homeless individuals often face barriers to receiving primary and preventative care, such as lack of access to health care services [5]. Consequently, homeless individuals are more likely to utilize the emergency department and undergo frequent hospitalizations relative to the general/domiciled population [6,7]. Research suggests that homeless individuals are three times more likely to have undergone emergency treatment in the hospital at least once in the past year relative to domiciled individuals [7]. In addition, homeless individuals have high rates of hospitalizations, with 17% reporting overnight hospitalization in the past year [8], compared to 7.1% of the domiciled population over the same time frame [9].

The elevated rates of emergency treatment and overnight hospitalization among homeless individuals may be explained by the high rates of acute and chronic diseases, traumatic injuries, and behavioral health conditions experienced among this population [7,10]. Further, the numerous problems experienced by homeless individuals are exacerbated by lack of resources to obtain medical care, such as insurance coverage and income [5]. Over 59% of homeless adults report being uninsured compared to 13.3% of domiciled adults [11,12]. Lack of health insurance can create barriers to care access and lead to high rates of emergency treatment and hospitalizations [5,13]. A study by Kushel et al. found that homeless individuals with medical insurance were more likely to have received ambulatory care (medical care at an outpatient site, excluding the emergency department) in the prior year and reported decreased barriers to care than those who were uninsured [7]. This suggests that having insurance may decrease homeless persons’ reliance on hospital-based or emergency department care. Sociodemographic factors such as differences in age, sex, education, and veteran status have also been found to be associated with disparities in overnight or emergency treatment among this population [5,13,14,15]. For example, one study found that older age was associated with emergency department (ED) visits among homeless persons [13], while another study found that younger homeless adults and those with less than a high school education were less likely to have a regular source of care [14]. However, other studies have not found an association between age or income and emergency treatment/hospitalizations [7,16,17].

Homeless women represent one of the fastest growing sub-populations among homeless individuals, with 3% more women experiencing homelessness in 2018 than in 2017, compared to 1% growth among men [1]. Although all homeless individuals experience various difficulties due to homelessness, the effects of health conditions and required care can differ significantly for women [15]. Homeless women have a unique array of medical needs, accompanied by serious comorbidities such as alcohol abuse, substance use, reproductive issues, victimization, physical assault, and chronic health problems [18,19,20]. Not surprisingly, homeless women have been found to be more likely than homeless men to report hospitalization or emergency treatment [15,21,22]. One study found that 43.9% of homeless women reported hospitalization in the past year, compared to 39.5% of homeless men [15]. Further, a greater percentage of homeless women reported frequent emergency room visits than homeless men (23.2% compared to 15.9%) [15].

Racial disparities also exist in emergency treatment or hospitalization among homeless individuals. One study found that African Americans were less likely than non-Latino whites to have had a hospitalization in the prior year [7]. Another found that African Americans were less likely to have made an ambulatory care visit than non-Latino Whites [17]. However, other studies have found no association between race and healthcare treatment among the homeless [23,24]. Previous research has also shown that veteran homeless individuals have more specific health needs and are at a high risk of poor health status compared to non-veteran homeless individuals [7,25,26]. For example, homeless veterans were found to report a higher number of medical problems [26] and were more likely to have had a hospitalization in the prior year than homeless non-veterans [7].

Finally, no studies have examined the associations between subjective social status and emergency treatment or hospital treatment among homeless individuals. However, it is included as a potential correlate of emergency room visits and hospitalizations in this study because it has been linked to numerous health risk behaviors among homeless individuals as shown in previous work conducted by members of the research team [27,28,29]. For example, previous studies have examined the association between subjective social status and health-related quality of life, and readiness to quit smoking, independently [27,28]. Further, examining the association between subjective social status and emergency and overnight treatment may reveal information that might otherwise not be captured using other socioeconomic indicators such as income or employment status, which has limited variability among the homeless population [30].

Overall, high rates of emergency and overnight treatment can lead to hospital overcrowding, which can reduce the overall quality and efficiency of care for both homeless and domiciled populations [15,21,22,31]. Better understanding these sociodemographic predictors and their associations with emergency treatment could help identify potential barriers to care among homeless populations or subgroups within them. While previous studies have examined factors that predict overnight and emergency treatment, in homeless samples, few include a host of predictors that were jointly examined. Thus, the current study seeks to elucidate the sociodemographic predictors of overnight and emergency treatment among a sample of homeless individuals in Dallas, Texas,

## 2. Materials and Methods

### 2.1. Participants and Procedures

Participants (N = 354) comprised homeless adults who were recruited in 2013 as part of a two-wave survey focused on health and health behaviors among homeless adults [32,33]. Recruitment was accomplished via flyers placed within a large shelter in Dallas, Texas. Eligibility criteria were: resided in shelter, >18 years of age, and had earned a score ≥4 on the Rapid Estimate of Adult Literacy in Medicine-Short Form (REALM-SF) indicating >6th grade English literacy level [33]. Interested participants consulted with shelter staff and were placed on a list with a time to report to an administrative area of the shelter for an informed consent process and data collection. Thus, a convenience sampling approach was used. Enrolled participants completed questionnaires on a laptop computer in a private area; each item was visible on the screen and read aloud to the participants via headphones. Remuneration for study participation entailed a $20 department store gift card. All procedures were approved by Institutional Review Boards of associated institutions.

### 2.2. Measures

#### 2.2.1. Predictors

Predictors examined included age, subjective social status, sex, race, education, health insurance, income, and veteran status. Age and subjective social status of participants were utilized as continuous variables. Subjective social status was measured using the community version of the MacArthur scale of subjective social status, which presents a 10-rung ladder (1 = bottom of the ladder to 10 = top of the ladder), that represents the scale on which a person indicates they occupy within a community [34]. Sex was categorized as male versus female while race was dichotomized as white versus black/other races, based on a low frequency of nonblack/nonwhite racial groups. Participants’ level of education was dichotomized as >high school degree versus ≤high school degree. Health insurance status was categorized as insured (any type of coverage) versus no insurance, yearly income as ≤$10,000 versus >$10,000, while veteran status was analyzed as yes or no.

#### 2.2.2. Outcome

The outcome variables were: (a) Stayed overnight for treatment in a hospital (yes or no); and (b) Treated in a hospital emergency room (yes or no), both over the past year.

#### 2.2.3. Analytic Plan

Descriptive statistics were used to examine the distribution of all variables. Logistic regression models were estimated for each outcome variable to assess the main effect of each predictor, entered jointly into the model. Individual assessment of these predictors has been utilized in previous studies examining different themes in predictors of emergency department visit and hospitalization. However, to incorporate a contrasting approach to our study, we decided to enter all the predictors jointly so that only the unique variance of each—accounting for the others—would be revealed. All analyses used two-tailed tests of significance with a statistical significance level set at alpha < 0.05. Data were analyzed using SAS version 9.4 (SAS Institute, Cary, NC, USA) [35].

## 3. Results

### 3.1. Participant Characteristics

Table 1 presents the frequencies for the variables of interest. Approximately 71% of study participants (N = 354; M_age_ = 44 ± 11) were men. The overall mean of subjective social status was 5.17 ± 2.5, while the racial makeup of the sample constituted 30% white and 70% black/other races. Most of the participants had a high school degree or higher (76%), were uninsured (77%), low-income (≤ $10,000 = 90%), and non-veterans (92%). About 28% and 53% respectively reported overnight treatment and emergency treatment.

### 3.2. Sociodemographic Predictors

Table 2 reports the results of adjusted logistic regression models for overnight and emergency treatment in hospital, with all predictors entered jointly into models. Sex emerged as the only predictor of overnight treatment in hospital (OR = 2.68, 95% CI = 1.61–4.47), and treatment in an emergency room (OR = 2.21, 95% CI = 1.34–3.65), such that women were more likely than men to be treated overnight and use emergency care.

## 4. Discussion

This study assessed the sociodemographic predictors of overnight and emergency hospital treatment among a sample of homeless adults. Consistent with prior research, homeless individuals in this sample underwent overnight and emergency hospital treatment at high rates relative to domiciled adults [36]. Overall, 53% of participants in our sample reported undergoing emergency treatment in a hospital in the past year. Comparatively, the Center for Disease Control and Prevention (CDC) reported 19% of domiciled adults underwent emergency treatment in the same time frame [20]. Also, 28% of participants in our sample reported being treated overnight in a hospital in the past year whereas only 7% of domiciled individuals underwent overnight treatment, as reported by the CDC [9]. Thus, findings emphasize the need for access to primary care and preventive services for homeless adults, which could result in decreased emergency room use and hospitalization. Lack of unmanaged acute symptoms could spark a chain of chronic conditions and difficulties resulting in hospitalization and emergency department use among homeless individuals.

In our study, sex was identified as a sociodemographic predictor of overnight and emergency treatment such that women were more likely than men to be treated overnight and use emergency care. These findings are consistent with previous research which found that homeless women were more likely than men to use the emergency department [15,21,22]. However, in the current study, all predictors were entered jointly into the model as opposed to individually assessing these predictors. In addition, results from one of the studies with similar findings focused solely on single homeless women [22], whereas our study encompassed all homeless women regardless of relationship status. Further, our sample relied on a convenience sample of individuals from one of the largest shelters in Dallas, Texas, serving more than 85% of the homeless individuals in the area [37], whereas most of the other studies were conducted outside the United States [21,22]. Overall, the current study highlights sex differences whereby women were more likely than men to be treated overnight and use emergency care. Future studies might benefit from stratified analyses by sex in order to establish potentially unique predictors of emergency room treatment for men and women, respectively. Unfortunately, small cell sizes for some variables precluded this approach—particularly for women—in the present study.

The differences by sex in overnight and emergency treatment may be explained by differences in perceived or actual health among men and women [38]. For example, previous research among domiciled populations has shown that women were more likely than men to rate their health as poor, which may lead to more frequent utilization of hospital care [38]. Further, homeless women have higher rates of severe mental illness and chronic medical illness, and similar rates of alcohol use behaviors when compared to homeless men [3,4,15,39]. Homeless women also lack access to preventive care such as prenatal care [40], mammograms, and Pap tests [41] and experience sexual assault at higher rates than homeless men [42], which may lead to increased need for emergency treatment. Given these results, there is a need for innovative interventions that address the primary behavioral, physical, and preventive health care needs of homeless women. These needs may include screening and treatment for substance abuse, mental health problems, and chronic disease [12,43]. Future studies should examine underlying risk factors among homeless women with high emergency department use and frequent hospitalization to further inform preventive efforts.

There are several limitations to this study. Due to the cross-sectional design of the study, causality cannot be inferred. Considering that participants were recruited from a single shelter in Dallas and largely comprised homeless men, results may not be generalizable and representative of all homeless adults and may lead to self-selection bias. However, more than 70% of the available sample in the shelter at the time were recruited and enrolled in the study. In addition, the shelter where recruitment took place was the third largest shelter in Dallas, Texas, serving more than 85% of the homeless individuals in the area [37]. Future studies could also utilize a more randomized recruitment approach to limit self-selection bias. Our sample also reflects the distribution of national homeless samples comprising mostly of men [44]. Future studies should consider replicating findings in additional homeless samples. Additionally, some sociodemographic factors that could predict emergency department visits and overnight hospitalization may not have been accounted for in this study because analyses were limited to variables collected in the parent study. However, we included predictors (e.g., subjective social status) which have not been previously explored in other studies. Another limitation is that we did not have information on the reasons for greater hospitalization and emergency department visits among women versus men, and if emergency department rates characterized appropriate use or if problems could have been resolved in non-emergency settings. However, an asset to this study is that, because all individuals were sheltered, seeking shelter was unlikely a reason for hospitalization. Another limitation is the nature of the predictor variables, which, in some cases, were limited by preordained, gross categories (e.g., income). It is possible that collection of these data in continuous form would have yielded different results. It is also notable that subjective social status was treated as a continuous variable in these analyses whereas it has been categorized in a handful of prior studies. Re-running analyses with subjective social status as a categorical predictor, however, yielded similar, null results. Finally, the self-reported nature of the outcome and predictor variables could have resulted in reporting bias. Future work might include medical record validations of hospitalization and emergency department use. Despite these limitations, the current work extends and explicates what is currently known about the predictors of emergency department use and hospitalization among individuals who are homeless.

## 5. Conclusions

Our results indicate that homeless women are more likely to utilize overnight and emergency treatment services than homeless men. Targeted interventions and policies are needed to address homeless women’s primary care needs to potentially reduce risk for costlier overnight hospitalization and emergency treatment. It is important to note, however, that although homeless men were less likely than women to use emergency treatment services, there is an expansive literature suggesting they too are at risk for adverse health outcomes and would also benefit from preventive and primary care services [45,46].

## Figures and Tables

**Table 1 ijerph-17-04271-t001:** Participant Characteristics among a Sample of Homeless Adults (N = 354).

Variable	Total N (%)/M [IQR]
Age	45 (5–52)
17 to 73 years old	
Subjective Social Status	5 (3–7)
Scale of 1 to 10	
Sex	252 (71.2)
Male	102 (28.8)
Female	
Race	
White	106 (29.9)
Black/other races	248 (70.1)
Education	86 (24.3)
<High school degree	268 (75.7)
≥High school degree	
Health Insurance	
Uninsured	272 (76.8)
Insured	82 (23.2)
Income (past year)	
≤$10,000	317 (89.5)
>$10,000	37 (10.5)
Veteran Status	
No	326 (92.1)
Yes	28 (7.9)
Overnight Treatment in Hospital (past year)	
No	254 (72)
Yes	100 (28)
Emergency Treatment in Hospital (past year)	
No	166 (46.9)
Yes	188 (53.1)

Note: M [IQR] = Median [Interquartile range, IQR].

**Table 2 ijerph-17-04271-t002:** Predictors of Overnight and Emergency Treatment in Hospital among a Sample of Homeless Adults (N = 354).

Variable	Overnight Treatment in Hospital	95% CI	Emergency Treatment in Hospital	95% CI
Age				
17–73 years old	1.01	0.99–1.03	1.01	0.99–1.03
Subjective Social Status				
Scale of 1 to 10	0.99	0.90–1.09	1.03	0.94–1.12
Sex				
Male	1	Referent	1	Referent
Female	2.68	1.61–4.47	2.21	1.34–3.65
Race				
Black/other races	1	Referent	1	Referent
White	1.05	0.62–1.77	1.36	0.84–2.21
Education				
<High school degree	1	Referent	1	Referent
≥High school degree	0.87	0.50–1.54	0.99	0.60–1.66
Health Insurance				
Uninsured	1	Referent	1	Referent
Insured	1.16	0.65–2.04	1.64	0.97–2.78
Income				
≤$10,000	1	Referent	1	Referent
>$10,000	1.52	0.72–3.21	1.05	0.52–2.13
Veteran Status				
No	1	Referent	1	Referent
Yes	0.64	0.23–1.77	0.84	0.38–1.88

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
