# Peer review of "Predictors of Overnight and Emergency Treatment among Homeless Adults"

_ijerph, 2020, doi:10.3390/ijerph17124271_

Round 1
Reviewer 1 Report
This article is well written but I do have one concern. In section 2.3.1 the authors state that they used the subjective social status as a continuous variable. This cannot be done since this variable is categorical and NOT continuous. For it to be continuous would mean that the distance from 1 to 2 is the same as the distance from 2 to 3 etc. Since this variable represents a classification based on a subjective value, it cannot be used as a continuous but only as a categorical variable. Therefore these analyses will need to be redone with the subjective social status classified as a categorical variable.
In addition, due to the small sample size, the authors should report the median and inter-quartile range (IQR) as an interval (q1, q3) for all of the continuous variables involved in the study. Also, Table 2 should make it clear that those results are based on the adjusted and not unadjusted analyses.
The conclusion needs to be expanded to men as well. What do your results tell you about men.men's behavior?
Reviewer 2 Report
This paper is well put together and professionally presented, but has a limited message, essentially that homeless women who volunteered to take part in a study on health risk factors are more likely than men in the same situation to have attended for emergency or overnight hospital care for any reason any time in the previous year. This is a finding that the authors report from previous papers in the introduction to the paper, so the findings of this study are not novel.
The study group are self-selecting, and while their representativeness is noted, as a key limitation to generalising from the study this needs more consideration.
The selection of underlying predictive factors is necessarily limited by the availability of dataset items, as the study is a secondary analysis of a data from a previous survey. The authors do provide justification for the selection of predictors sex, race and selective social status, but selective social status is justified only via previous work from members of this team. I would like to see this made clear in the text.
The authors state that a strength of their study is to examine a range of risk factors 'in tandem'. It is not clear what they mean by this. In the model, the authors state that all predictors were jointly entered into the model, which raises the question of why a model was used at all. The authors are clear about this (line 166) but don't explain why individual assessment of the relationship of each factor with hospital use followed by modelling (if indicated) of key relationships was not used. Could they explain why they made this decision. Given that you already knew that women used the ED more than men, could you add to what is known by looking at the other predictors separately for men and women to try and unpick this relationship in your study group?
The stated aim of the study is to 'inform interventions to promote lower hospitalisation and emergency treatment rates among this population through increasing the accessibility of preventive services to the most at risk subgroups'. This implies that identifying at risk subgroups is key. However the study does not do this. The conclusions need to reflect this.
Round 2
Reviewer 1 Report
The authors answered my comments satisfactorily.
Reviewer 2 Report
The authors have responded with clarity and precision to all the points raised in my original review. I am happy with the manuscript as it stands.